# The Representation of Children’s Participation in Guidelines for Planning and Designing Public Playspaces: A Scoping Review with “Best Fit” Framework Synthesis

**DOI:** 10.3390/ijerph20105823

**Published:** 2023-05-15

**Authors:** Rianne Jansens, Maria Prellwitz, Alexandra Olofsson, Helen Lynch

**Affiliations:** 1Department of Health, Education and Technology, Luleå University of Technology, 971 87 Luleå, Sweden; maria.prellwitz@ltu.se (M.P.); alexandra.olofsson@ltu.se (A.O.); 2Department of Occupational Science and Occupational Therapy, University College Cork, T12 AK54 Cork, Ireland; h.lynch@ucc.ie

**Keywords:** children’s rights, design by inclusion, playgrounds, policy, policy implementation

## Abstract

For children, meaningful participation in community life includes being able to access places for play. Such community playspaces are potentially important for all children, including those with disabilities. Yet, children are rarely asked for their views on the design of playspaces, which can further contribute to exclusionary practices and undermine children’s rights to share their views on matters that affect them. In this scoping review, we aim to analyze guidelines and identify strategies for supporting children’s participation rights when planning public playspaces. Guidelines are practical tools used by local policymakers when creating community playspaces, which are important sites for children’s outdoor play. In total, forty-two guidelines were identified that addressed children’s participation rights, along with community involvement. Qualitative evidence synthesis with a “best fit” framework approach was used, informed by Lundy’s model of children’s participation. The findings revealed the importance of initial community involvement as a critical prerequisite. Strategies for children’s participation mostly concerned “*space* and *voice*” (for children of diverse abilities), with little attention paid to giving their views due weight. This evidence shows that there is a significant gap in knowledge surrounding policy development and implementation to support adults and children to cooperate equally in designing playspaces. Future directions for research in children’s participation require a focus on combined community–children participation approaches in public playspace design. Such work could strengthen and facilitate the role of adults as bearers of the duty to implement the rights of children. This review generated inclusive strategies in planning public playspaces, which could support local policymakers in this complex multi-layered process.

## 1. Introduction

Play parks and playgrounds are typically accessible to the public and can be defined as playspaces, which is “a term often used to describe an adult-designed and designated place for playing” [1] (p. 118). Public playspaces are provided in many countries because play is acknowledged as a fundamental occupation and concern for children and consequently, an important societal concern [2]. In recent years, this provision has extended to inclusive approaches to design by considering the diverse play needs of children, including those with disabilities [3,4,5]. From a children’s rights perspective, play is a freely chosen self-directed and intrinsically motivated activity, process or behavior and is acknowledged as one of children’s rights [6,7,8]. Yet, children’s rights to have a say in matters that affect them are not commonly considered when designing playspaces. Therefore, it is imperative to develop playspaces that are in accordance with the play priorities of diverse children and adopt children’s participation rights-based approach to planning and design [5]. In this paper, the term “children of diverse abilities” refers to the range of capacities, strengths and resources of all children, including those with disabilities. In addition, the term “children” is used to address all children under 18 years of age, as defined by the United Nations’ Convention of the Rights of the Child (UNCRC), unless other synonymous terms are used by the authors to refer to this age group (such as young people).

Children value parks and playgrounds that contain a wide range of affordances for play, including natural and artificial features and private spots that can provide opportunities for social activities and play [9,10,11,12]. However, public parks and playgrounds have typically reflected adults’ views on controlling children’s spatial behaviors and adults’ priorities for children’s play, for example, promoting physical play in jungle gym playgrounds [1]. Yet, there are differences between adults’ and children’s priorities for play, especially in outdoor playspaces. For example, research has identified that in some contexts, children’s needs for opportunities to manipulate physical environments can differ from the views of local park workers [10], children with disabilities often want to play away from their carers [13] and there are differences in risk perceptions between children and parents [14]. Therefore, it is essential to identify children’s priorities for play and allow children to be listened to, as well as adults, when planning playspaces.

Children’s participation rights are a group of rights within the UNCRC that facilitate children under 18 years of age to share their views on matters that affect them and ensure that their views are given due weight [15]. Many tools and models have been developed to support the implementation of these participation rights [16]. Among them, the most commonly used model to guide children’s participation in decision-making processes is Lundy’s model [17]. In Lundy’s model, children’s participation processes are represented as having four facets, mirroring the two key parts of article 12: *space* and *voice* (the right to express their views on matters that affect them) and *audience* and *influence* (the right to have their views given due weight) [17]. These four facets represent a process sequence that has the intended outcome in the last facet *“influence”* [17,18]. The facet of *space* is about creating opportunities for participation through the purposeful encouragement of children’s involvement, i.e., asking them about matters affecting them, as well as giving children opportunities not to be involved and creating safe and inclusive environments for participation [17]. The facet of *voice* is about providing multiple means of expression that fit children’s capacity to express their views freely by providing sufficient time, giving access to child-friendly information and supporting the capacity of adults to facilitate this process [17]. The facet of *audience* is about requiring adults to engage and listen actively, as well as providing children with secure opportunities to share their views with people who are responsible. It is also about informing children of when and how their views can be communicated with decision-makers and giving them the chance to be part of this communication process [17]. The facet of *influence* is about ensuring that children are informed about what has happened with their views and how and why their views have or have not influenced decisions [17].

While the relevance of children’s rights to play and to participation is visible in some policies, there is little guidance on how to bring both into practice [19,20]. In addition, while a review of evidence for children’s participation has been conducted recently [11], the use of such evidence in practice remains limited [5]. In the absence of general guidance, local policymakers tend to rely on playspace guidelines for implementing practice rather than policy or research evidence [21]. Guidelines are defined as “sources providing information intended to advise on how something should be done or what something should be” [22]. Many guidelines exist for designing playspaces that aim to provide procedural guidance for local authorities, including needs assessment, planning, design, community engagement, funding and the realization of play facilities. However, these guidelines are often unclear about implementing children’s participation approaches, even though they are the main users of these playspaces and, as noted earlier, experts on play. To summarize, there is a lack of evidence on how children’s participation in playspace planning and design is represented, if at all. Therefore, the purpose of this review was to reveal insights that could support local policymakers in facilitating the participation of all children in playspace provision, irrespective of their abilities. The aim of this study was to identify children’s participation strategies (i.e., concepts, principles, approaches, plans, sources and gaps) in guidelines for designing public playspaces.

This review addressed the following questions: how guidelines for designing public playspaces consider children’s participation and what strategies are evident or are missing for ensuring children’s participation rights are enacted. 

## 2. Materials and Methods

Given the nature of the review questions, a scoping review method was selected to conduct the guideline review, following the Johanna Briggs Institute guidelines to ensure methodological rigor [23,24,25]. This scoping review was of the gray literature, which is defined as literature/documents that are not typical research papers and, therefore, are accessible outside of traditional sites, such as libraries or databases [25]. The research protocol was published on an open-access digital platform [26]. 

### 2.1. Search Strategies

The search for relevant sources involved four stages: (1) a search of the web-based search engine Google, using the search words “guidelines design play space”, (2) a search of the resource sections of the websites of organizations advocating for children and play, (3) asking members of two international NGOs on outdoor play and designing play facilities about their sources and (4) the citation mining of the records of the previous search results and known scientific articles. Appendix A describes the search strategies in detail and presents a description of the identified documents. The Google search, the consultation of the websites of relevant organizations and the questioning of the two international organizations were carried out between 19 October and 30 November 2021. The citation mining was conducted in the following weeks. 

### 2.2. Eligibility

English language sources were included for review when they described the design of a public playspace and included children’s participation (or an aspect of it according to the UNCRC or a concept that could include children’s participation, such as community involvement and stakeholder consultation). Sources describing the design of playgrounds in educational or daycare centers were excluded because they are not always accessible to citizens. Books and book chapters were excluded because municipal design teams are unlikely to use them and the content or focus of such material is more diverse compared to guidelines. No limitations were set for the publication year. No information was available about the development of guidelines for designing public playspaces. Time limitations were not used in order to maximize the inclusion of diverse material. 

### 2.3. Selection Process

In total, 76 guidelines for planning and designing public playspaces were fully screened by two reviewers (the first author and one of the other authors), which resulted in the selection of 42 guidelines (Figure 1). Some organizations published multiple guidelines, all of which were included because they were presented as new guidelines, so an analysis seemed justified. A detailed description of the screening and final selection processes is presented in Appendix A.

### 2.4. Critical Appraisal of Included Sources

Efforts were made to evaluate the quality of the included guidelines using the assessing unconventional evidence (ACE) tool, which supports the assessment of the quality of different types of policy-relevant evidence [27]. The ACE tool encompasses 11 criteria regarding purpose and context, the completeness or accuracy of the information presented, the extent to which evidence supports the information and various aspects of reporting, such as rights and ethics [27]. This provided a means of assessing the strengths and limitations of the included guidelines. In total, seven guidelines had moderate or minor concerns because they used empirical data to support guideline content but did not have clear links and lacked some methodological information. The methodological quality of 35 sources was found to be poor, with serious limitations concerning the description of objectives, intended audience and rationale, which were presented without any supporting evidence. 

### 2.5. Data Extraction and Synthesis

Generic data were extracted by the first author using a tool developed by the research team, based on a Johanna Briggs Institute template [24]. Data on the research topic, i.e., children’s participation, were mapped in two phases of data extraction using the “best fit” framework synthesis, a qualitative evidence synthesis method that allows researchers to conduct studies with a priori frameworks and also confirm, critique or add to the frameworks [28,29]. For phase one, the researcher was required to identify a suitable framework for data extraction and analysis. A literature review of the theories and models of children’s participation was conducted to identify a framework for children’s participation. In total, 6 scientific and 24 non-scientific sources were examined for suitability, and Lundy’s model of children’s participation [17] was the most frequently used approach. The planning checklist of the national participation framework of Ireland applies Lundy’s model, using 23 questions to analyze children’s participation in practice contexts, and was considered suitable for the framework synthesis [30]. This provided the research team with a tool for extraction. Data from the guidelines were analyzed, mapped and synthesized to this children’s participation framework of 23 items. In phase two, “leftover data” were extracted and analyzed using interpretivist thematic analysis as this is considered good practice for “best fit” framework synthesis [29]. This analytic approach was used to synthesize data on children’s participation, community involvement and stakeholder consultation. Two researchers (R.J. and H.L.) conducted the qualitative evidence synthesis using NVivo software [31]. 

## 3. Results

### 3.1. Characteristics of the Included Guidelines on Designing Public Playspaces

The term “guideline” was dominant over other terms, such as briefing, planning framework or toolkit, as shown by the names of the 42 included sources. The size of the documents ranged from 6 to 156 pages. Their structures differed from leaflets with key principles to process descriptions with or without checklists, e.g., addressing planning and design aspects, the relevance of inclusive play and the participation of different stakeholders. Guidelines were developed by government agencies (17), non-governmental organizations (16), playground industries (7) and an interprofessional stakeholder group (2). The majority of guidelines originated from the UK (12), followed by Australia (10), the USA (9), Canada (3), Ireland, (2), New Zealand (1) and Hong Kong (1), with two guidelines originating from two cooperating countries (Australia–Thailand and USA–Canada). The biggest group of guidelines was published in the past decade, with 25 guidelines being published since 2010. The objectives of the guidelines were to assist local authorities in planning processes and highlight best practice. The intended audiences were community members, social and planning professionals and, sometimes, children and young people. Some guidelines had a particular emphasis, such as nature-based play, inclusive playspaces or children’s and young people’s participation. In total, 15 out of the 42 guidelines were specifically aimed at creating playspaces that are accessible and inclusive for children of diverse abilities.

### 3.2. Modes for Participation

The included guidelines described different ways or “modes” of participation: collaborations that propose activities at different points in processes by consulting children a few times, working with a children’s advisory group or having children participate in the design team (19) or consultations and, thus, one-off events in the design process (17). No guidelines described child-led participation. 

Table 1 provides further descriptive information on the 42 guidelines. (Appendix A provides for additional information to Table 1 such as authors, topics addressed, special focus, reference to UNCRC, assessment with ACE tool and URL).

### 3.3. “Best Fit” Framework Synthesis: Themes on Children’s Participation and Community Involvement

Overall, seven themes were identified: (1) giving space and time for consulting with the local community, (2) identifying the needs of the community, beyond play, through an active, meaningful and empowered approach, (3) establishing a shared vision responsive to community’s needs, (4) giving children safe, inclusive opportunities to form and express their views about playspaces, (5) facilitating children to express their views, (6) informing children who will be listening to their views on playspaces and (7) informing children of actions taken as a result of their shared views.

Themes four to seven reflect the four facets of Lundy’s model of children’s participation (i.e., *space, voice, audience* and *influence*), which served as the framework for synthesizing the data. However, since 35% of the codes could not be allocated to Lundy’s model, three new themes emerged that addressed community involvement. The importance of consulting not only children but first and foremost with the local community is mirrored in the descriptions of themes one to three. The community involvement themes were considered features of a process that was an extension of Lundy’s model of children’s participation, which need to come first in the process as community engagement is required before engaging with children (this process is depicted in Figure 2; also see Appendix A for an overview of strategies for community involvement and children’s participation represented in guidelines for designing public playspaces).

#### 3.3.1. Theme 1: Giving Space and Time for Consulting with the Local Community

This theme reflects the importance of the identification of relevant stakeholders in the community and forming networks for the first phase of the consultation process. Various guidelines described the importance of first involving community members in the consultation for designing public playspaces.

The main aim of this phase was to establish a collaborative intersectoral process with all relevant stakeholders from the start. This included a diverse range of people, such as adults and elderly people living in the area, their representative forums, professionals working with children, local entrepreneurs and elected officers. This theme explored six key strategies relating to engagement with stakeholders: (1) consulting community members, (2) consulting users, local children and young people, (3) consulting children with disabilities and their caregivers, (4) consulting professionals, (5) early engagement from the start and (6) regular consultations throughout the process.

The first step that was most commonly outlined was a strategy for consulting adults as they were the gatekeepers for creating playspaces, and their experiences and opinions for the spaces needed to be identified from the outset. Identifying relevant stakeholders was also a key part of this strategy, for example, the chairs of local organizations, intermediaries for hard-to-reach people and caregivers. Children and, in some guidelines, young people were considered key stakeholders in planning and designing playspaces; however, communication with adult stakeholders was prioritized. Two guidelines stressed the principle of the early involvement of the community, and that communication with stakeholders should be tailored to their interests. Five guidelines described strategies for carrying out regular community involvement throughout the planning, design, construction, maintenance and progressive enhancement phases. A key principle throughout this early engagement and involvement process was the need for interactive characters in meetings, which was proposed as important for the overall goal of achieving better used playspaces and enjoyable neighborhoods where children are seen and heard as members of the community.

Overall, several guidelines described many strategies for implementing initial phases for children’s participation, which involved prioritizing the provision of space and time for adult community members as the playspaces were to be seen as community spaces, and adults essentially were the gatekeepers for children’s participation. This “space and time” phase included engagement with multiple key stakeholders concerned with the diverse aspects of playspace planning and provision that therefore warranted diverse approaches to maximize community engagement. 

#### 3.3.2. Theme 2: Identifying the Needs of the Community, beyond Play, through an Active, Meaningful and Empowered Approach

This theme explains how community involvement should ideally take place. The included guidelines described many strategies for informing the community about the planning and design process and community engagement strategies. The aim of this phase was to generate collective knowledge about community members’ needs and priorities. This theme explored four strategies: (1) providing information about projects and processes, (2) providing site visits with informal engagements and discussions, (3) connecting with users and community members through intermediaries, local forums and families and (4) applying active and strength-based approaches with community members, including children and young people.

Listening to stakeholders and identifying community needs required broad perspectives, i.e., the current use of the place, nearby services and access to them, potential resources, the needs and wishes of the community and on-site discussions. Three guidelines described ways that local forums, voluntary groups and professionals could mediate to find and engage with members of the local community. Most importantly, the community involvement process should be characterized as an active, realistic, meaningful and empowering process with various formal and informal activities with the community, including children and young people, which was represented in 19 guidelines. Various strategies were proposed to achieve this, such as identifying underutilized spaces, observing children in playspaces, discussions on what kinds of experiences adults would like children to have, enhancing the understanding of how the community already support children and needs identified by adults. This process of community engagement needed to be transparent and required regular communication about the project and consultation process with the broader community. A key strategy documented for maximizing the success of this phase of community involvement was the need for a facilitator who could connect, communicate and lead the co-production of knowledge regarding the playspace and their neighborhoods. It was proposed that this person would also play a key role in managerial aspects, such as establishing strategies for community and child involvement, integrating projects into inventories of all parks, playgrounds and open spaces, identifying geographical, demographic and transport data for analyzing play facilities and the neighborhood and constructing a steering committee with the appointment of people responsible for children’s participation.

Overall, according to these guidelines, community engagement needed to be characterized by various informal and formal strategies, both in the community and on the sites. Such strategies required leadership and facilitators to communicate effectively and regularly with the diverse stakeholders so that all stakeholders understood the experiences and perceptions supporting children’s play. The goal was to gather and obtain consensus on the needs and priorities of the whole community, which was a broader issue than just gathering information on children’s play needs.

#### 3.3.3. Theme 3: Establishing a Shared Vision Responsive to Community’s Needs

In this theme, the need for addressing aspirations and goals during the early phases of community involvement is addressed. The aim of this phase of community involvement was to provide rationales and describe the possible benefits and outcomes of recruiting and engaging with community members. This theme explored three strategies: (1) acknowledging children’s expertise, (2) aiming for community ownership and reductions in the risk of vandalism and (3) achieving outcomes that are responsive to community needs and aspirations.

In total, 23 of the 42 guidelines addressed reasons why community involvement was considered important as they included a focus on the core values that underpinned their approaches. For example, it was proposed that by including children’s participation through community involvement, playspaces could be better designed and used. Moreover, guidelines advocated for such core values as they could optimize the effectiveness and utilization of playspaces and broader neighborhoods. Varied rationales were given for community involvement; for example, it could lead to greater tolerance for outdoor play and contribute to practicing the ownership and better maintenance of spaces, thereby possibly reducing vandalism. Additionally, 11 guidelines proposed that meaningful community involvement could ensure that community needs were met and that relationships with local authorities could be beneficial for future projects. Community involvement together with children’s participation was viewed and valued as a process that could contribute to creating livable, enjoyable and sustainable environments for all. In relation to children, it was about viewing all children with diverse abilities and children from disadvantaged backgrounds as members of the community through a number of strategies, such as acknowledging their play expertise, accepting that children have different perspectives on playspaces than adults, valuing their agency and supporting their learning about decision-making processes.

Overall, a key characteristic of these guidelines was describing the aspirations and intended goals of community involvement, such as optimizing the effectiveness and use of playspaces and wider neighborhoods, as well as building relationships with and within the community where children were recognized as members. 

#### 3.3.4. Theme 4: Giving Children Safe, Inclusive Opportunities to Form and Express Their Views about Playspaces

This theme represents the first facet of Lundy’s model, which addresses the implications of article 12, i.e., providing *space* for children not only to express their views but also form them. The aim of this phase is to provide guidance for adults to create safe and inclusive opportunities for children to support them in participating in the decision-making process for designing playspaces. This theme explored six strategies: (1) the early involvement of children, (2) creating sustainable involvement, (3) ensuring inclusive and accessible processes, (4) involving children who are affected, (5) supporting children to feel safe and comfortable expressing themselves and (6) being able to provide support for children when they become upset. Giving children safe and inclusive opportunities also meant giving children space and time to form and express their views.

In total, 25 of the 42 guidelines proposed that children should be consulted as users of public playspaces, while 17 took general approaches to community engagement. Additionally, 12 guidelines described the need for sustainable involvement through ongoing consultation processes that demonstrated dialogue and mutual learning, as well as through validation with other children in same age group, professionals and general community members. The strategies for ensuring inclusive and accessible processes included setting up participation activities at strategic locations, seeking out the views of underrepresented groups of children and including a variety of people in terms of age, ability, gender, socioeconomic status, race and culture. Moreover, 19 guidelines specifically emphasized that children with disabilities should be given the opportunity to form and express their opinions about playspaces, with the suggestion of also seeking guidance from parents and professional caregivers. Experiential learning, including reflection and discussion, could offer one approach for supporting children to feel safe and comfortable in forming and expressing their views. Another strategy was to ensure access to a knowledgeable facilitator who was open and had time to engage with individuals and groups of children in a variety of activities. While the framework established the need to provide support for children should they become upset, no guidelines addressed this consideration.

Overall, most guidelines emphasized the need for space for children to form and express their opinions, and their focus was mostly on engaging children in the design of public playspaces and facilitating their sustained involvement from the beginning. However, the guidelines overlooked certain prerequisites, such as safety, inclusive processes and support for when children became upset.

#### 3.3.5. Theme 5: Facilitating Children to Express Their Views

This theme refers to *voice*, i.e., the second facet of Lundy’s model that relates to children’s right to express their views freely. This phase aims to support adults in maximizing children’s capabilities to share their views. This theme explored six strategies: (1) ensuring there is a list of topics on which you want to hear children’s views, (2) ensuring that the key focus of the process stays on topic, (3) informing children that participation is voluntary at all times, (4) supporting children in giving their own views, (5) ensuring a range of ways for children to express themselves and (6) allowing children to identify topics to discuss and to add to the list of topics. Among these six items, items one, two and three were about preparing children (prerequisites), while items four, five and six were about facilitating the process of expressing themselves freely.

This part of the framework identified the most strategies for the four facets of Lundy’s model of children’s participation. In total, 23 guidelines described different recommendations and examples of ways to support children to share their views and suggested a variety of activities and methods, such as drawing, modeling, photography or video, storytelling, visual or sensory mapping, small group discussions and site visits with games or discussions. The plans and strategies outlined a range of activities and ways for adapting them for the ages of the children and for accompanying information to be accessible to all children involved. The facilitating adult needed to consider which activities and approaches would best suit the children and actual contexts. Some guidelines recommended questions to pose to children about their play, such as where, with whom or what and when they play. During this process, the facilitator was expected or advised to avoid questions about play equipment and to provide extra support for children with additional needs. Furthermore, nine guidelines identified topics for gathering children’s views, and three guidelines identified the need to give children the opportunity to identify topics for the participation process and the need to let children develop themes and ideas that needed to be considered for creating lists of priorities. While the framework established the need to ensure that children knew about the voluntary nature of participation, no guidelines described how children should be informed about the voluntary character of participation nor that they could withdraw at any time.

Overall, the included guidelines focused primarily on age-appropriate child-friendly methods and activities for facilitating children to express their views, with little emphasis on the prerequisites for doing so. 

#### 3.3.6. Theme 6: Informing Children Who Will Be Listening to Their Views on Playspaces

This theme specifies the *audience* facet of Lundy’s model, which is about children’s right to have their views listened to by people involved in decision-making. This phase of children’s participation aims to encourage adults to communicate with children about what will happen with their shared views and inform them about the decision-making process. This theme explored five strategies: (1) informing children about to whom, how and when their views will be communicated, (2) showing commitment to being informed and influenced by their views, (3) informing children about the identification of decision-makers, (4) reporting back to children about the decision-making process in child-friendly ways, (5) giving children the opportunity to confirm their views and (6) giving children a role in communicating their views.

In total, five guidelines described considerations for informing children about to whom, how and when their views were to be communicated. Several strategies were proposed, such as being clear that the children were not the designers of the playspaces and explaining to children that their views were to be combined with other data, such as historical information, local development plans and community involvement. Moreover, six guidelines addressed community commitment to children being informed and influenced by their views. This mostly addressed the *audience* facet of children’s participation by showing that there was commitment to being informed and influenced by their views. Examples of these strategies included informing children that their views were to be collected in reports and handed over to professionals (i.e., designers, steering groups, etc.) and going back to participating children with draft designs. Only one guideline recommended providing child-friendly versions of the collated views. Strategies were proposed for more effectively involving diverse children in communication processes about their views. These included working with children to create visual tools about what and how they inform decision-makers and the broader community. However, informing children about the identification and involvement of decision-makers was not described in the guidelines. Likewise, there were no suggestions or examples for how to help children to verify the accuracy of the recording of their views.

Overall, the included guidelines rarely described strategies for informing children about who will listen to their views and the associated processes, such as the verification of their shared views, the identification of decision-makers and reports on giving their views weight.

#### 3.3.7. Theme 7: Informing Children of Actions Taken as a Result of Their Shared Views

The last theme is about ensuring that children’s views are given due weight in accordance with their age and capacity and reflected the *influence* facet of Lundy’s model. This phase aims to provide adults with guidance on communication with children about how and to what extent their views influenced the decision-making process. This theme explored five strategies: (1) informing children about the scope of their influence, (2) giving age-appropriate feedback during design processes, (3) planning to make sure that children’s views impact decisions, (4) giving children age-appropriated feedback on how their views are used and (5) providing opportunities to evaluate the participation process. 

In total, 11 guidelines described strategies for informing children about the scope of their influence, for example, informing children about the parameters of their involvement, giving a checklist to guide the process and informing children about the overall aim. Strategies were also proposed for informing children about the plans to ensure their views would impact decisions. This involved describing the role of the designer in this process, for example, presenting a concept design in child-friendly ways. Another strategy was to involve a local council to support the participatory design process and for contractors to allow children to help with planting. Some guidelines proposed that children should be given regular (age-appropriate and accessible) updates at key points during the whole process. Guidelines also suggested other strategies, such as explaining how children’s views were used and the reasons for decisions made. These strategies stressed the need to explain to children that all suggestions would be considered but not everything would be feasible and why particular aspects would or would not be feasible. Only two guidelines described strategies for evaluating the participation process through questionnaires, focus groups or continual on-site reviews of the playspace and the participation process.

Overall, informing children about the actions taken as a result of their shared views mainly aimed at informing them about the possibilities and limitations of influencing decision-making. Providing age-appropriate feedback or the opportunity to evaluate the participation process was only described in a few guidelines. 

## 4. Discussion

This scoping review aimed to analyze guidelines for planning and designing public playspaces in order to identify different strategies for children’s participation. In total, 42 guidelines were identified that addressed children’s participation, including 19 that focused specifically on the participation rights of children with disabilities, along with other key issues, such as community involvement or stakeholder consultation, which demonstrated the importance placed on the participation rights of all community members when designing public playspaces. The amount and depth of participation recommendations varied widely by source as there were different types of affiliated organizations, including NGOs, government agencies and the playground industry; however, all of them reflected complex and multilayered processes involving professionals and volunteers from diverse backgrounds, including children and adults as community members. Overall, our analysis identified seven themes that provided a process framework or pathway for strategies for community engagement and children’s participation in public playspace design.

From the framework synthesis, four themes were derived from Lundy’s model of children’s participation. The findings showed that while Lundy’s model was a widely accepted process framework for operationalizing children’s participation, guidelines for designing public playspaces mainly emphasized strategies that offered children the opportunity to express their views, i.e., strategies reflecting the facets of *space* and *voice*. However, little attention was paid to children’s right to have their views be given due weight in accordance with their age and ability, which referred to the facets of *audience* and *influence* from Lundy’s model. Lundy noted in her model that adults’ understanding of children’s capacities can create some tensions as adults may consider that they know best for children. However, according to Lundy (2007) and Tisdall (2017), article 12 clearly states that adults, as duty bearers, need to do everything possible to enable participation, including the participation of disadvantaged children, such as children with disabilities. This requires the avoidance of well-known problems, such as adults’ lack of awareness, taking on managerial approaches and not fully understanding the implications of children’s rights [32,33]. In the absence of recommendations for adults as duty bearers in guidelines for planning and designing public playspaces on how to act on children’s views, the question may arise whether these guidelines represent children’s rights-based approach in the true sense of the word. Consequently, the empowerment of children and their ability to experience and influence democratic processes on matters that affect them in their community is unlikely to be fully realized. To address this gap, there is a need to take the time to build open and genuine relationships in which children are respected as citizens and experts in their own lives, combined with mobilizing local champions for the effective implementation of children’s rights-based participation approach [34,35,36]. The need for more attention to be paid to children’s right to have their views given due weight has been echoed in diverse examples of children’s participation in policy, practice and research [34]. The findings in this review could add to such work and highlighted that more focused attention is needed for children’s participation in the area of playspace development and for being more mindful of underrepresented groups, such as children with disabilities, since play and participation rights are for all children.

This review also highlighted new aspects for consideration when applying Lundy’s model of children’s participation to designing public playspaces. There is a need to first provide the space and time to involve the local community and seek their commitment through the active, meaningful and empowering on-site mapping of community needs, beyond the focus of play. Playspaces are considered community spaces, and, therefore, information on the current use of spaces and any desired changes should be gathered from the perspective of all stakeholders. Moreover, community residents, business leaders, teachers, youth workers and urban planners can facilitate or hinder children’s participation process. To address this effectively, as noted in theme three, the mindsets of adults in the local community and those of involved professionals are considered crucial to bridging the gap between the children’s world and that of adults and avoiding adultism (i.e., when children and young people are considered inferior in a participation process) [37]. Changing adult mindsets is possible and has been confirmed in some studies, which have found changes in attitude among adults when children were participating in playspace design [11,38]. Strategies for communicating rationales and the benefits of engaging community members could help to evoke community aspirations for creating better used and maintained playspaces, as well as pleasant neighborhoods where all children with diverse abilities are valued as community members. Playspaces are not fixed or neutral places but are instead lively environments that are interwoven with relationships [1,39]. Therefore, to integrate children into the public realm of enjoyable neighborhoods, the importance of paying attention to mindset is crucial in order to take into account not only the physical but also the sociocultural aspects of spaces from the perspectives of all citizens [39].

With regard to future directions for children’s participation, as noted earlier, the identification and synthesis of seven themes provides a process framework or pathway for community involvement (i.e., the first three themes) preceding children’s participation when designing public playspaces. This framework could be considered as a complement to Lundy’s model, as shown in Figure 2. However, the results of this scoping review did not explain how community involvement and children’s participation could best co-exist while taking into account different and possibly competing interests when planning and designing public playspaces. Parental civic beliefs, civic participation and socialization practices greatly influence whether children participate in decision-making processes and how they operationalize their civic participation [40]. Therefore, future work is needed in policy development and implementation to support the adoption of strategies and approaches that help adults and children to cooperate equally in re-occurring meetings and events when designing playspaces. Such work could strengthen and facilitate the roles of adults as bearers of the duty to implement children’s rights [18]. This qualitative evidence synthesis demonstrated the co-existing importance of community involvement and children’s participation for local play provision and emphasized the need for opportunities to create child-led designs of playspaces that acknowledge children, regardless their abilities or backgrounds, as the main users of the spaces and as play experts.

According to our analysis, there are no standards or clarity on topics that should be included in guidelines for designing public playspaces, which is a gap that needs to be addressed to facilitate local authority play provision. Government organizations and NGOs try to facilitate the operationalization of children’s rights through such sources, yet research-based recommendations to improve planning guidelines are scarce [41]. The results of this scoping review could help local policymakers to implement two important children’s rights in everyday community life. The combination of the right to participation and the right to play is not yet visible in tools for operationalizing policy, such as the included guidelines for designing playspaces, which could be due to its complexity and the required intersectoral cooperation. However, as Davey and Lundy clearly stated, “The UNCRC is the hub of a wheel and if any of the spokes were to break, the wheel would buckle” [20] (p. 11). Thus, combining and integrating different children’s rights is imperative, as stated in the UNCRC itself, and creating participation opportunities in community playspace design could be a logical first step [42]. The key contributions of this study could inform best practice in policy implementation and influence future directions in children’s participation.

However, there were several limitations to this study. Although efforts were made to conduct this scoping review as systematically and rigorously as possible, its methodological flaws need to be acknowledged. Searching the gray literature using a web-based search engine was influenced by unknown algorithms and could not provide an exact overview of the intended sources. The assumption that playspaces in (pre)school settings are for private use resulted in the exclusion of guidelines for designing school playgrounds. Limiting the search to English language sources meant that guidelines for playspaces in other languages were missed. To our knowledge, the “best fit” framework synthesis has not yet been used with guidelines as the sources of qualitative evidence. Accordingly, the ACE tool was used to ensure quality assessment, and we found that this tool could guide efforts to strengthen the rather weak methodological quality of these non-conventional sources [27].

This scoping review identified the need for further research into how children’s participation and play can be integrated and operationalized in the local community, how the quality of children’s participation can be pursued and, most importantly, how all children can be represented (as some groups are underrepresented in decision-making processes, such as children with disabilities) [11,43,44,45,46]. This kind of implementation research is essential for informing policy development as there is a need for child-centered evidence-based practice in urban planning and a need to make children’s rights-based approach less dependent on individual commitments in complex policy processes [19,41].

## 5. Conclusions

This scoping review identified 42 guidelines for designing public playspaces that addressed children’s participation, community involvement or stakeholder consultation. A qualitative evidence synthesis using a “best fit” framework approach demonstrated that children’s right to share their views (i.e., the facets of *space* and *voice* from Lundy’s model of children’s participation) was addressed most in varied strategies. However, children’s right to have their views be given due weight (i.e., the facets of *audience* and *influence* from Lundy’s model) received little attention; therefore, children’s potential learning and their influence on the decision-making process regarding community play provision should be questioned. Additionally, while some guidelines included the need to consider children with disabilities, this was an underrepresented area of concern overall. The seven identified themes provided a process framework or pathway for community involvement preceding children’s participation when designing public playspaces. Giving space and time to first involve adult community members in a meaningful way in order to understand their needs and aspirations was considered important as playspaces should be seen as community spaces and adults are essentially key gatekeepers for children’s participation.

## Figures and Tables

**Figure 1 ijerph-20-05823-f001:**
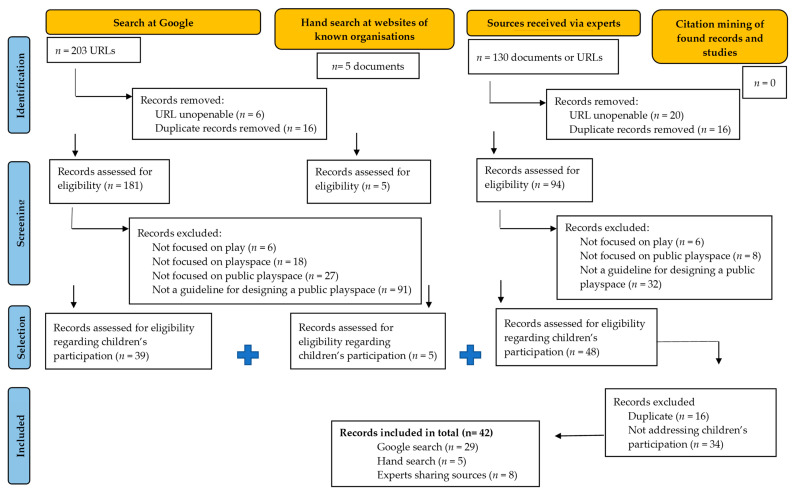
Prisma flow diagram of identification, screening, and selection of guidelines for designing a public playspace representing children’s participation.

**Figure 2 ijerph-20-05823-f002:**
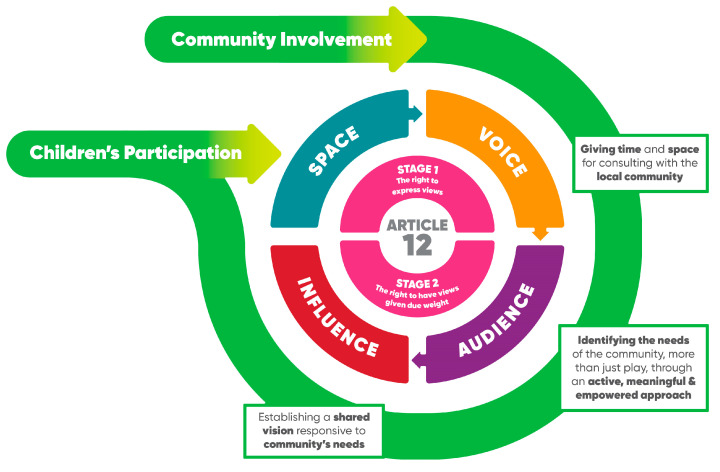
Lundy’s model of children’s participation, extended to community involvement for designing public playspaces. Reprinted/adapted with permission from Participation Framework. National Framework for Children and Young People’s Participation in Decision-Making. 2021. Department of Children, Equality, Disability, Integration and Youth [30] and Prof. Lundy [17]).

**Table 1 ijerph-20-05823-t001:** A descriptive summary of guidelines for designing public playspaces that address children’s participation.

No.	Affiliated Institution or Organization	Type of Organization	Year of Publication	Country	Title and Subtitle	Type of Document According to Authors	Objective of the Guideline According to Authors	Intended Audience According to Authors	Modus of Participation
1	Australian Heart Foundation	Non-governmental organization (NGO)	2013	Australia	Space for active play. Developing child-inspired play space for older children	Guideline	To assist local governments in undertaking “healthy urban planning”	Local governments	Consultation
2	CABE	Cooperation of diverse stakeholders	2008	United Kingdom	Designing and planning for play	Briefing	To highlight best practice in design and strategies and encourage the greater use of creative and natural playspaces	Local planners, developers and architects	Consultation
3	Christopher and Dana Reeve Foundation	NGO	n.d.	USA	Toolkit for building an inclusive community playground	Toolkit	To provide community advocates with resources and tips to broaden their understanding of the requirements of inclusive playgrounds and provide suggestions to facilitate fundraising efforts	Community advocates	Collaboration
4	City of Ballarat	Government agency	2014	Australia	City of Ballarat play space planning framework	Planning framework	To provide a planning framework to improve and develop playspaces	Citizens, professionals and others involved in the processes	Collaboration
5	Creo	Playground building industry	n.d.	New Zealand	We create smart public play spaces	Not described	Not described	Clients	Too little information
6	Denver Parks and Recreation	Government agency	2017	USA–Canada	Nature play design guidelines	Guideline	To provide a framework for parks and recreation, urban drainage, forestry and public works. To establish unstructured sensory play, align the nature play design process and develop a maintenance and facilities process	Public servants, funders, health and wellness advocates, communities and children	Collaboration
7	DESSA	Government agency	2007	Ireland	Play for all. Providing play facilities for disabled children	Publication	To support community development projects, family resource centers and other community development organizations in ensuring their play facilities are accessible and welcoming to all children	Community development organizations, planners, architects, local authority staff and interested individuals	Consultation
8	Free Play Network	NGO network	2008	United Kingdom	Design for play: A guide to creating successful play spaces	Presentation	To support the creation of successful playspaces	Not described	Consultation
9	Geelong Australia	Government agency	2012	Australia	Geelong play strategy: Part 2. Planning and design guidelines, management, marketing and maintenance of play space	Report	To provide a good overview of playground development considerations	The community of the Greater City of Geelong	Collaboration
10	Government South Australia	Government agency	n.d.	Australia	Inclusive play guidelines for accessible play spaces (easy read version)	Guideline	To provide an easy read guideline	Anyone planning or building new playgrounds or playspaces	Consultation
11	Greater London Authority	Government agency	2012	UK	Shaping neighborhoods: play and informal recreation. Supplementary planning guidance	Planning guidance	To support how planning should be carried out, with practical advice, in particular by negotiating for enough playspace to be set aside in new developments	Not described	Consultation
12	Hags	Playground building industry	2019	Worldwide	Inclusive play design guide	Guide	To contribute to more inclusive spaces for everyone	Individuals and groups aiming to create playspaces in their communities	Collaboration
13	HNH (Healthy New Hampshire) Foundation and NRPC (Nashua Regional Planning Commission)	NGO	2017	USA	Planning for play. A parks and playground guidebook for New Hampshire	Guidebook	To understand the process of park and playground development, from planning to implementation	Local authorities in New Hampshire, USA	Consultation
14	Illinois Department of Natural Resources	Government agency	2004	USA	A guide to playground planning	Guide	To provide information and assistance in the planning, design, installation and maintenance of public playgrounds	Local governmental agencies with minimal or no permanent staff, as well as community groups with limited knowledge or experience in developing public playgrounds	Consultation
15	Inclusive SA (South Australia)	Government agency	n.d.	Australia	Guidelines for accessible play spaces	Guideline	To challenge standard practice and inspire innovative design solutions that ensure playspaces can be enjoyed by every South Australian	Local governments, schools, early childhood learning centers, design professionals and others	Consultation
16	Inspiring Scotland, Play Scotland and the Nancy Ovens Trust	NGO network	2018	Scotland, UK	Free to play guide to accessible and inclusive play spaces	Guide	To assist any group that come together to develop or improve public playspaces	Friends of parks, community councils, community planning partnerships and groups of local parents, carers, professionals and youngsters	Consultation
17	Landcom	Government agency	2008	Australia	Open space design guideline	Guidelines	To help to deliver the best outcomes for open spaces	The two principal partners and the end owner (usually this means local councils and Landcom development staff)	Collaboration
18	Landscape Structures Inc.	Playground building industry	2018	USA	Inclusive play space design planning guide	Guide	To help to create inclusive playgrounds that are unique to their communities	Not described	Collaboration
19	National Playing Fields Association	NGO	2004	UK	Can play, will play. Disabled children and access to outdoor playgrounds	Report	To advise local authorities and other playground managers and assist them in meeting the requirements of the Disability Discrimination Act	Local authorities and other playground managers	Collaboration
20	NCB (National Children’s Bureau)	NGO	2009	UK	How to involve children and young people in designing and developing play spaces	guide	To be used alongside the Design for Play: A guide to creating	All those involved in designing and developing playspaces for children and young people	Collaboration
21	NSW (New South Wales) Government	Government agency	2019	Australia	Everyone can play guideline. A guideline to create inclusive play spaces	Guideline	To provide a key resource for the planning, design and evaluation of new and existing playspaces in NSW (New South Wales, Australia)	Councils, community leaders, landscape architects and local residents	Consultation
22	Office of Deputy Prime Minister	Government agency	2003	UK	Developing Accessible Play Space. A Good Practice Guide	Guide	To advise on developing accessible playspaces that disabled children can use	All stakeholders	Collaboration
23	Play England, Department for Children, Schools and Families, Department for Culture, Media and Sport	NGO	2008	England, UK	Design for Play. A guide to creating successful place spaces	Guide (non-statutory guidance)	To support good practice in the development and improvement of public playspaces	Commissioners, designers, playbuilders, and local authorities	Consultation
24	Play Wales	NGO	2012	Wales, UK	Play spaces—planning and design	Not described	Not described	Not described	Consultation
25	Play Wales	NGO	2016	Wales, UK	Community toolkit. Developing and managing play spaces	Toolkit	To provide a single source of support and signposting for community groups to help them to navigate some of the challenges of managing or developing playspaces	Anyone taking responsibility for managing or developing playspaces in communities	Collaboration
26	Play Wales	NGO	2021	Wales, UK	Developing and managing play spaces. Community toolkit	Toolkit (providing guidance and tools)	To provide a single source of support and signposting to navigate some of the challenges of managing or developing playspaces	Anyone taking responsibility for managing or developing playspaces in communities, e.g., community councils, local play associations or resident groups	Collaboration
27	Playcore	Playground building industry	2012	USA	Blueprint for Play Design It.	Toolkit	To inspire communities to maximize the play design process for community-based initiatives	Not described	Determine the level of involvement
28	Playground Ideas	NGO	n.d.	Australia–Thailand	5 steps for a better place to play	Manual	To empower people to go out and create amazing playspaces with their communities	Not described	Collaboration
29	Playright	NGO	2016	Hong Kong	Inclusive Play Space Guide	Guide	To advise and inspire the design of accessible and inclusive playspaces in Hong Kong	Designers, play providers and operators of unsupervised playspaces in Hong Kong	Collaboration
30	Playworld	Playground building industry	2015	USA	Inclusive Play Design Guide	Guide	To guide the creation of great outdoor play environments for everyone	People who care about inclusion and aim to create playspaces in their communities	Collaboration
31	Playworld Systems	Playground building industry	2015	USA	Playground 101 Guide. How to build a playground in 10 easy steps	Guide	To help answer questions, as well as provide educational resources	Not described	Collaboration
32	Playworld Systems	Playground building industry	2019	USA	Inclusive Play Design Guide	Guide	To offer inspiration and guidance to support the design of inclusive and universally designed outdoor playgrounds	Landscape architects, park and recreation staff, municipal employees, parent/teacher groups, community groups, parents and educators	Collaboration
33	Real Play Coalition	Cooperation of diverse stakeholders	2020	Worldwide	Reclaiming Play in Cities. The Real Play Coalition Approach	Publication	To share the initial steps toward developing an urban play framework (a holistic tool for facilitating play)	City stakeholders, including decision-makers, urban practitioners and investors	Too little information
34	Rick Hansen Foundation	NGO	n.d.	Canada	Let’s play toolkit. Creating inclusive play spaces for children of all abilities	Toolkit	To provide information and best practices for designing accessible playspaces for all children	Communities	Consultation
35	Rick Hansen Foundation	NGO	2020	Canada	A guide to creating accessible play spaces	Guide/toolkit	To support the design of accessible and inclusive playspaces	Communities	Consultation
36	State of Victoria, Dept for Victorian Communities	Government agency	2007	Australia	The good play space guide: “I can play too”	Guide	To examine the reasons why playspaces can limit access to some children and identify how improvements can be made to increase participation by all children in play	The providers of public playspaces	Collaboration
37	Touched by Olivia	NGO	n.d.	Australia	The principles for inclusive play	Principles	Not described	Not described	Collaboration
38	Tualatin Hills Park and Recreation District	Government agency	2012	USA	Nature play area guidelines	Guidelines/document	To support the design and implementation of nature play areas	Tualatin Hills Park and Recreation District staff and contractors	Too little information
39	Unknown	Government agency	2014	Canada	Integrated accessibility standards regulation guidelines. Part 4.1: Design of public spaces standard	Guideline/standard	To inform about the regulations for outdoor spaces	Organizations interested in constructing or redeveloping outdoor spaces	Consultation
40	Waverley Council	Government agency	2021	Australia	Inclusive play space study report Abridged version	Report/study	To provide practical guidance on inclusive playspace design and help to translate best practice policy into actionable principles	Inclusive play specialists, landscape architects and other interested parties	Collaboration
41	Wexford County Council Community Development Department	Government agency	2018	Ireland	Developing a play area in your community. A step-by-step guide	Guide/booklet	To help to develop play areas for children in communities	Communities	Consultation
42	Wokingham Borough Council	Government agency	2018	UK	Play space design guide	Guide	To provide clients, developers and designers with guidance and specific requirements for the design of playspaces within the borough	Planning officers	Too little information

Legenda: Type of organization: non-governmental organization (NGO); government agency; playground building industry; a cooperation of diverse stakeholders. Modes of participation: consultation; collaboration; child-led participation.

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
