# Peer review of "The Representation of Children’s Participation in Guidelines for Planning and Designing Public Playspaces: A Scoping Review with “Best Fit” Framework Synthesis"

_ijerph, 2023, doi:10.3390/ijerph20105823_

Round 1

Reviewer 1 Report

No policy implementation and future direction is given. Describe limitation of the study  and contribution to the next level.

Reviewer 2 Report

The purpose of this review is to reveal insights to support local policymakers in facilitating the participation of all children and identify strategies to ensure children’s participation rights are enacted. This review also highlighted adultism and community engagement in achieving children's participation. The conclusions are significant, and here are some partial modification suggestions:

  1. The abstract should include a section on the conclusion.
  2. Regarding section 2.1. Search strategies: some of the documents collected by the two non-governmental organization personnel are poorly described. For example, what types of documents are they? Where do they come from? How did they obtain them? In addition, why does the search scope not include books? Why are publication years not limited?
  3. Please define "grey literature".
  4. The entire article discusses children, but the term "children and young people" is frequently used in the text (such as lines 199 and 262), making the research subjects unclear. To be precise, there are differences in the needs of children and young people. Has the author considered the differences in designing guidelines for the two groups?
  5. Sections related to children in Frameworks 1-3 overlap with the Lundy model, such as consulting with local children, consulting with children (lines 226-227, etc.), and the content of Framework 5: Facilitating children to express their views. Frameworks 1-3 should focus more on emphasizing the communication process with adults in the early stages.

Reviewer 3 Report

Dear authors, 

Thank you for your contribution. The work is well developed and structured. Involving children in the development of their public play spaces is such an interesting topic. Nevertheless, below you can find few comments to improve the manuscript.

·      Did the authors also consider academic works/publications in the search? It is not specified in section 2.1. If yes, please clarify. If not, I would suggest including these types of works in the search.  

 ·      It would be interesting to show in the manuscript at least one real example of play space generated with the contribution of children. 

·      Table 1 is difficult to read. I would suggest the authors try to compact it. Also, it would help show the headings in every page. Summarising somehow the content of the table after it would help too.

Thank you for your attention. 

Reviewer 4 Report

This paper is concerned about the children playspace because it is considered as one of the social concerns to create user friendly environment while accommodating the needs of diverse children including the disabled ones. As per the UN mandate, having a user-friendly play space is considered as one of the children’s rights to demonstrate their diverse talents inherent on them which might have been hindered due to their physical conditions. Thus, it is imperative that a play place should be designed to address the concerns of children while taking their opinions, however, such opinions need to be controlled and regulated by practical approach while dovetailing real world experiences from adult guardians. This article summarizes a series of grey literature that are prepared in accordance with the guidelines developed by the Johanna Briggs Institute following Lundy model which suggests respecting both space and voice. All literature resources published in English language were searched using Google Engine between 19 October and 30 November 2021 and ideas were incorporated from guardians and expert groups from a few NGOs.

Of the 76 guidelines developed by the Assessing UN-Conventional Evidence (ACE) encompassing 11 criteria regarding the children’s public spaces, forty-two rules and regulations (guidelines) regarding children’s participation were found relevant while planning and designing public play spaces. Seven guidelines had moderate or minor concerns because they used empirical data to support guideline content but without clear links and lacked some methodological information. Most guidelines originated in the UK, Australia, USA, Canada, Ireland, New Zealand, Hong Kong, with two guidelines from two cooperating countries (Australia-Thailand and USA-Canada) are referred here in this article.

The paper has listed seven themes related to children play spaces. These include: 1) giving space and time for consulting with the local community, 2) identifying the needs of the community, beyond play, through an active, meaningful, and empowered approach, 3) establishing a shared vision responsive to community’s needs, 4) giving children and young people safe, inclusive opportunities to form and express their views about playspaces, 5) facilitating children to express their views, 6) informing children who will be listening to their views on playspaces, and  7) informing children of actions taken as a result of their shared view.

My take:

This paper as such does not contribute any original ideas from the authors to the scientific communities because it is merely a grey literature review based. However, I see the possibilities that policy makers and planners may find this document relevant to quickly glance what is needed for the all-round development of diverse children from a play space. It may help policy makers and planners to give a consideration of children’s spaces which are seriously missing in the rapidly urbanizing world that has converted many spaces into concrete jungle without maintaining friendly amenities to children.

Author Response

Please see attachment. As you suggested, we consulted English proof reading service. 
